# TFP Bioeconomy Impact post Covid-19 on the agricultural economy

**C. A. Zuniga-Gonzalez**  *

Agricultural and Veterinary Sciences Faculty, Agroecology Department, Research Centre of Bioeconomy and Climate Change, National Autonomous University of Nicaragua, Leon, Nicaragua

* czuniga@ct.unanleon.edu.ni

**Data Availability Statement:** Figshare: Data for: TFP Bioeconomy Impact post Covid-19 on agricultural economy.csv. figshare. Dataset. https://doi.org/10.6084/m9.figshare.22337914.

## Abstract

### Background

This research was focused on measuring the TFP bioeconomy post-Covid-19 in six regions of the world.

### Methods

The panel data was organized with FAO Statistics data. Linear programming with an enveloping data analysis (DEA) approach was used to measure the Malmquist TFP indices to determine the inter-annual productivity and technical efficiency changes by region.

### Results

The results show that the effect of Covid-19 on the bioeconomy productivity during the period 2012–2021 on average decreased by 11.6%. This effect was explained by the decomposition of the productivity change into the changes in technical efficiency. The workers decreased their efficiency by 11.7%. In the Northern American region, it decreased by 21.6%, in the Southern European region by 10.1, and in Western Europe by 11.7%.

### Conclusion

The results show a downward trend that was affected in the year 2019 by Covid-19, however, it was possible to recover in the following year. One of the conclusions of these results is the effect of the immediate strategies that the governments of the region implemented. This effect was a little slower in the North American, Southeastern, and Eastern European regions. Finally, it is concluded that the measures implemented by the governments in the studied regions had an increasing effect in conditions of variable scale returns. In other words, the companies that remained on a constant scale decreased.

**Funding:** The author received no specific funding for this work. The funders had no role in study design, data collection and analysis, decision to publish, or preparation of the manuscript.

**Competing interests:** The author have declared that no competing interests exist.

## I. Introduction

The world economy has experienced the effects of Covid-19, unleashing the biggest crisis in more than a century. Some authors have pointed out that one of the effects is inequality between countries [1] that is to say, that the recovery after the Covid impasse will be gradual and uneven. Emerging economies and vulnerable groups will take much longer to recover from the income and livelihoods lost to the pandemic [1,2].

Deepening inequalities within and between states, short-term governments depend on the crisis responses, and emerging threats to an equitable recovery outlined. These are some of the economic impacts of the pandemic and emerging risks to recovery [3].

Solow [4] was who contributed to establishing the total factor of productivity as an operational concept, based on the production function. In his article "Technical change and the aggregate production function", published in 1957, he describes a way of separating the variations in output per capita due to technical change and the availability of capital per capita.

Approximately in 2007, the European Union created the FP7 platform, which is an economic reactivation program where the bioeconomy is defined as an economic activity that involves the use of biotechnology and biomass in the production of goods, services or energy [5]. The terms are widely used by regional development agencies, national and international organizations, and biotechnology companies. It is precisely from this year that scientists begin to apply Total Factor Productivity as a tool to measure productivity and technical efficiency in bio based economies.

Fuente [6] aimed to provide a management tool to help mussel aquaculture practices farmers identify optimal culture strategies and use production inputs efficiently. Obi and Visser [7] researched the data envelopment analysis (DEA)-Malmquist non-parametric frontier technique for the measurement of plantation forest harvesting operations in New Zealand over 10 years (2009–2018).

Fritsche *et al.*, [8] comment that several initiatives related to bioeconomic development began before the COVID-19 pandemic. They are add that the health crisis coincided with the EU's political agenda shift towards bio-economic development.

There have been post-Covid-19 efforts to measure the change in total factor productivity in different regions of the world (see Table 1). Part of the research efforts to measure TFP is the works of Hao *et al.* [9] that focused on the DEA-Malmquist index and entropy method to measure the manufacturing green total factor productivity (GTFP) and the level of digital economy level from 2011 to 2018, respectively. Additionally, measured the effects of talent aggregation and financial scale adjustments using the generalized method model in the digital economy. Wang *et al.*, [10] provide a reference for the green development of countries and regions, emphasizing the importance of green development policies adapting to local conditions and

**Table 1. Methods used for Malmquist index for bioeconomic.**

| Methods | Measurement | Cited by |
|---|---|---|
| (DEA)-Malmquist non-parametric | Plantation forest harvesting operations | Obi and Visser [7] |
| TFP Malmquist index | Measurement Aquaculture | Fuentes [6] |
| DEA-Malmquist index and entropy method (GTFP) | Manufacturing green total factor productivity | Hao et al. [9] |
| TFP | Green development policies | Wang et al., [10] |
| GTFP | Digital economy | Liu et al. [11] |
| ER, GF, FDI, and IGT | Green productivity | Tong et al. [12] |
| SBM model to measure GTFEE | Market segment index | Zhou *et al.* [13] |

time and providing evidence for market-oriented green economy development. Liu *et al.* [11], consider that the digital economy can significantly improve China's GTFP; however, there are clear regional differences. The digital economy was related to the GTFP index to relate the promotion of the growth of the economy in the city through improvements in industrial structures. Tong *et al.* [12], integrate ER, GF, FDI, and IGT into a coherent framework of green productivity and considers the negative yield in GTFP as ignored in the previous ones. In environmental planning, these empirical findings have implications for decision-makers. Zhou *et al.* [13] establish an SBM model to measure GTFEE by using considering undesirable outputs, and environmental supervision indicator systems as unexpected output, and then using the price-relative price method to calculate the market segment index. Next, this author points out that he used a basic linear regression model to analyze the effect of corruption and market segmentation, identifying it with GTFEE considering China as the object of study (see Table 1).

This study aims to develop a management tool that allows decision makers in the field of economic policy in the agricultural sector to identify optimal investment strategies in innovation in bio based economies and use production inputs efficiently.

For this purpose, it has used the Malmquist Indices methodological tool to evaluate the performance of the different region post-Covid economic policy strategies.

It apply non-parametric frontier analysis to determine the impact of the measures taken to deal with the negative effects of Covid 19. It estimates the non-parametric Malmquist indices for analyzing the change in productivity throughout the post-Covid 19 investment period to determine productivity and technical efficiency by country and region.

This article is divided into 5 sections. Section 2 briefly introduces the literature review of bioeconomy, Total Factor Productivity, and the scale efficiency measurement concepts developed by Farrel [14] and other authors. Section 3 outlines how the DEA methodology using linear programming methods (Malmquist-DEA) may be empirically implemented and describes made from FAO-STAT. Section 4 describes the results and discussion of the processing panel data. Finally, concluding points are made in Section 5.

## II. Literature review

In this section two main topics are addressed, one is bioeconomy as the economic theory that the European Union has been promoting since 2007 and the second is total factor productivity, although this tool was applied in 1957 [4], it is in the years 2007–2023 that is innovatively applied in the bioeconomy.

### 2.1 Bioeconomy

The VOSviewer software (RRID: SCR_023516) was used in the literature review. 9,372 documents were analyzed, of which only 61 established the limit to highlight the topic of bioeconomy during and post Covif-19. Fig 1 shows the main authors' map highlighting the bioeconomy and Covid topic. It denotes four clusters where the pandemic has resulted in delays in implementing climate policies and altered priorities from climate action (Vo. T.P.T *et al.*, [15]); Agarwal *et al.* [16] highlight the role of various cyanobacterial species that are a source of green and clean energy along with their high potential in the production of biodegradable plastics.

The bioeconomy approach is developed unevenly in the regions of the world. In Europe in the past decade, it has been developed through the Horizon 2020 program, while in the Americas region only Canada, the United States, Brazil, and Argentina have adopted the bioeconomy as a development axis in their government agendas [17]. Patermann, & Aguilar [18] briefly

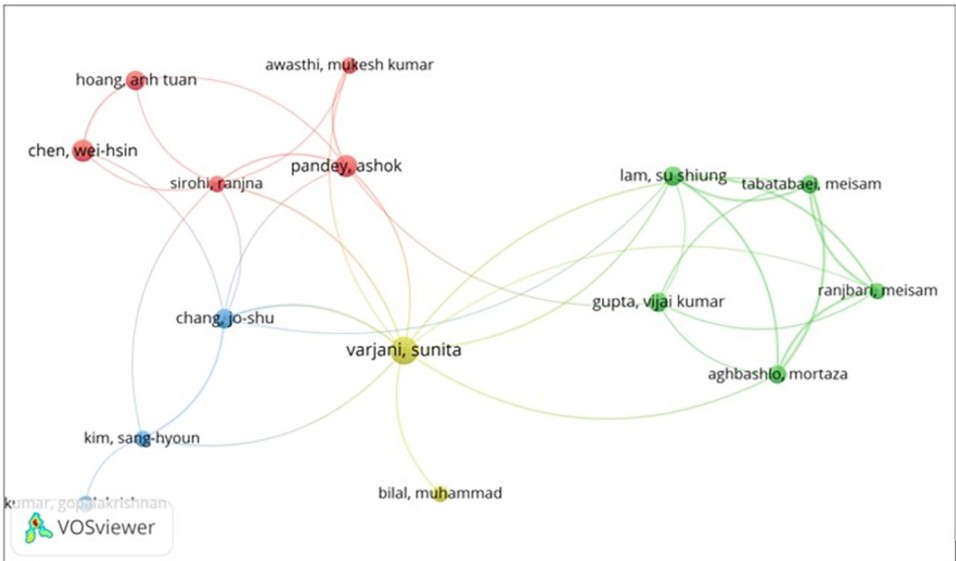

**Fig 1. Map based on co-citation Dimension data for Bioeconomy during and post Covid-19.**

analyzed the two most important impacts of the EU Strategy on the Bioeconomy. These were the Bioeconomy within the Horizon 2020 Program (2014–2020) and the creation of a public-private association of bioindustries. Fig 2 highlights the regions that contribute to the

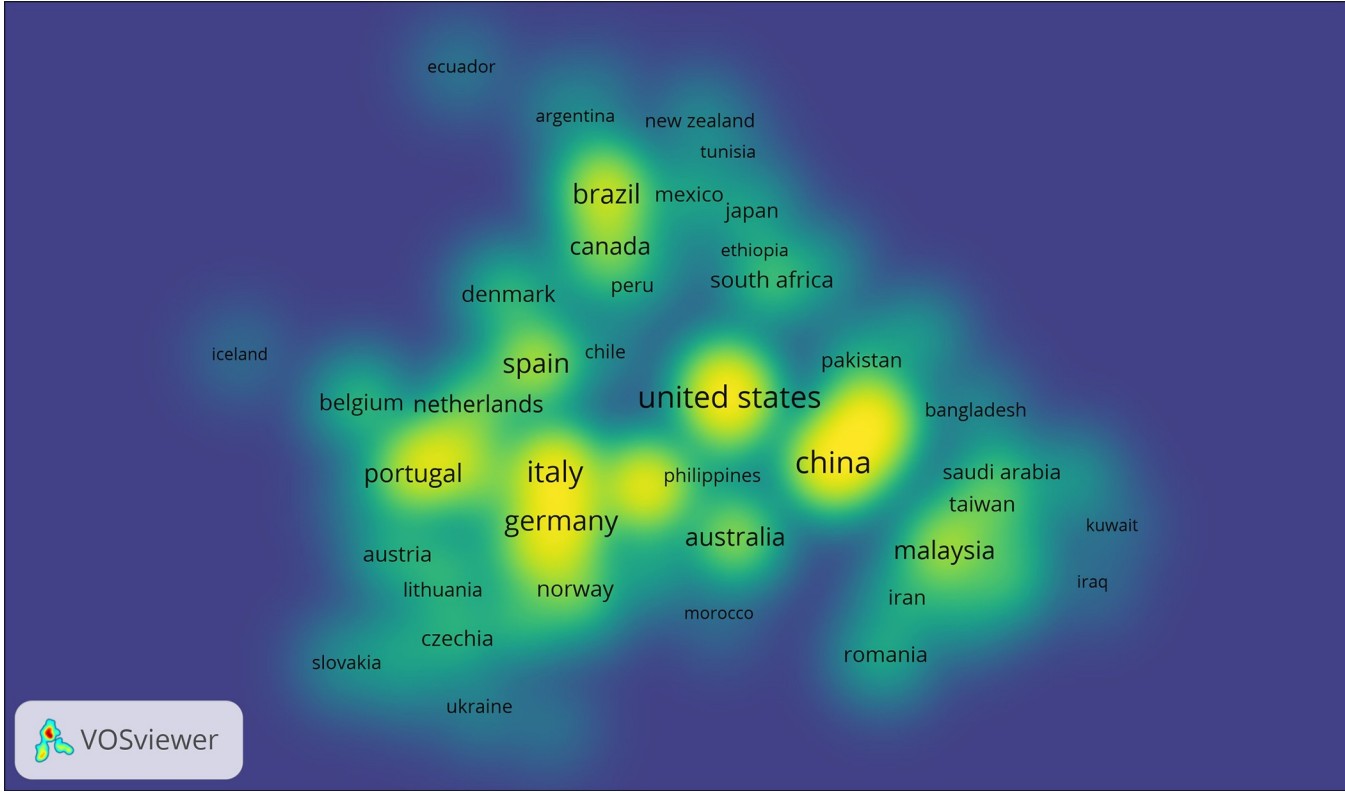

**Fig 2. Map of countries that contribute to Bioeconomy and Covid topic 2012–2021.**

bioeconomy during and post Covid-19 topic. In Northern America is the United States, and in South America is Brazil. In Northern Europe is the United Kingdom, and the Southern is Italy and Germany.

Galanakis *et al*. [19], explore how the bioeconomy can enhance the resilience and sustainability of bio-based, food, and energy systems in post-COVID-19. These authors consider variables such as technological innovation, rural economies, tourism, food, biocities, ecosystem services, and the environment, others identify them as productive paths of the bioeconomy [20]. They also highlight the importance of integrating other variables such as the fashion industry, arts, and culture.

Sarkar *et al*. [21] highlight some of the key policy drivers on an overarching national scale and those specific to agricultural research and innovation that are critical to fostering a supportive environment for innovation and a sustainable bioeconomy. They add that Canadian agriculture is facing the challenges of climate change, sustainable agriculture, clean technologies, and agricultural productivity.

Woźniak & Tyczewska [22] present the challenges and threats during and after the COVID-19 pandemic, as well as opportunities that can be brought by for bioeconomy development.

In the review of the literature it was possible to observe that the bioeconomy has been developing unevenly, since the European Union adopted it as an alternative for this century, for its part in the American region it has been gradually adopted.

In this situation, scientists have the challenge of measuring and quantifying the economies so that decision-makers consider their economic policy (investment) based on productivity and efficiency in conventional economies or in conditions with bioeconomy or bio-based economics.

## 2.2 Total factor productivity

As previously mentioned, the Malmquist indices tool has been developed in order to apply it to the different facets that the bioeconomy has been developing. For this purpose, it will use the controlled variables indicated in section III.

Färe *et al*. [23] proposed an output-based Malmquist productivity change and technical efficiency change.

$$\psi_0\big(\varrho_{t+1}, \rho_{t+1}, \varrho_t, \rho_t\big) = \left[\frac{\delta_0^t(\rho_{t+1}, \varrho_{t+1})}{\delta_0^t(\rho_t, \varrho_t)} \times \frac{\delta_0^{t+1}(\rho_{t+1}, \varrho_{t+1})}{\delta_0^{t+1}(\rho_t, \varrho_t)}\right]^{\frac{1}{2}} \qquad \text{(LP1)}$$

$LP_1$ represents the productivity at the point $(p_{t+1}, \varrho_{t+1})$ relative to the production point $(p_t, \varrho_t)$. In such a way that it considers values from 0 to 1 for measuring productivity. It will understand that there is a growth of the TFP when the value is 1 from period t to period t+1. To estimate $LP_1$, the four functions of the distances of the components must be calculated, of which the LP problems are involved (similar to those conducted to calculate the Farrel (1957), measure in technical efficiency (TE).

Assuming CRS technology. The oriented LP CRS output used to compute $\delta_0^t(\rho_t, \varrho_t)$ is defined in $LP_{1, 2}$, however, the constraint on convexity (VRS) has been removed and the subscription time is included. This is:

$$\left[\delta_0^t(\rho_t, \varrho_t)\right]^{-1} = max_{\phi,\lambda}\phi,$$

$$s.a - \phi\rho_{it} + Y_t\lambda \geq 0,$$

$$\rho_{it} - P_t\lambda \geq 0,$$

$$\lambda \geq 0, \tag{LP2}$$

The LP problems are a simple variation of this:

$$[\delta_0^t(\rho_t, \varrho_t]^{-1} = max_{\phi,\lambda}\phi,$$

$$s.a - \phi\varrho_{i,t+1} + Y_{t+1}\lambda \geq 0,$$

$$\rho_{i,t+1} - P_{t+1}\lambda \geq 0,$$

$$\lambda \geq 0, \tag{LP3}$$

$$[\delta_0^t(\rho_{t+1}, \varrho_{t+1}]^{-1} = max_{\phi,\lambda}\phi,$$

$$s.a - \phi\varrho_{i,t+1} + Y_t\lambda \geq 0,$$

$$\rho_{it} - P_t\lambda \geq 0,$$

$$\lambda \geq 0, \tag{LP4}$$

$$[\delta_0^{t+1}(\rho_t, \varrho_t]^{-1} = max_{\phi,\lambda}\phi,$$

$$s.a - \phi\varrho_{it} + Y_t\lambda \geq 0,$$

$$\rho_{it} - P_{t+1}\lambda \geq 0,$$

$$\lambda \geq 0, \tag{LP5}$$

Note that LP $_{4, 5}$ production points are compared with different period-type technologies, the parameter $\phi$ does not need to be $\geq 1$, as when calculating the Farrell (1957) efficiency. The points must be below the production amount allowed.

This is most likely to occur at LP $_4$, where the period t-1 production point is associated with the technology with period t. With technological advances, values of $\phi < 1$ are possible. Note that if a tech comeback happens, it could happen in PL $_5$ as well, but it's unlikely.

A few things to notice are that $\phi$, and $\lambda$ are likely to take different values within four LPs. Also, note that all four LPs must be computed for each region in the sample. Also note that if you add a period, you will need to calculate 3 PLs per region (to create the correction rate). If we are measuring the T period, we need to calculate the (3T-2) LP for each region in the sample. So for N Region = 6, we need to compute N * (3T-2) PL. This study with N = 6 regions and T = 10 time periods (2012–2021) should provide 6 * (3*10–2) = 168 LPs.

## 2.3 Scale efficiency

The approach of Färe et al. (1994) can be extended by decomposing (CRS) Technical Efficiency Change (tech) into Scale Efficiency and "pure" Change VRS (sech, psech) Technical Efficiency

components. This includes calculating two additional LPs (when comparing the economic output of two regions). These involve repeating LP $_{2, 3}$ for each additional convexity constraint (N1'$\lambda$ = 1). That is, it computes the distance function associated with the VRS technology (not the CRS technology). Using the CRS and VRS values, we can calculate the residue-free scale efficiency effect using the method above.

## III. Methodology and panel data

This article used the application of Malmquist DEA methods to panel data to calculate the indices of total factor productivity (TFP) change; technological change; technical efficiency change and scale efficiency [23]. DEAP 2.1, Data Envelopment Analysis (Computer) Program (RRID: SCR_023002) was used, Coelli [24]. Three text files were used for running computing. The text file refers to panel data containing 60 observations of six regions over the 2012–2021 years period. The second file is Instructions, where the procedure is indicated, and the third is the results (output) that are shown in the results sections. One output is considered with five inputs listed in the next section.

Applying LP 1, output-based Malmquist productivity change and technical efficiency.

$$\psi_0\left(VA_{t+1}, LU_{t+1}, VA_t, LU_t\right) = \left[\frac{\delta_0^t(LU_{t+1}, VA_{t+1})}{\delta_0^t(LU_t, VA_t)} \times \frac{\delta_0^{t+1}(LU_{t+1}, VA_{t+1})}{\delta_0^{t+1}(LU_t, VA_t)}\right]^{\frac{1}{2}} \tag{LP6}$$

Then for each input:

$$\psi_0\left(VA_{t+1}, UCS_{t+1}, VA_t, UCS_t\right) = \left[\frac{\delta_0^t(UCS_{t+1}, VA_{t+1})}{\delta_0^t(UCS_t, VA_t)} \times \frac{\delta_0^{t+1}(UCS_{t+1}, VA_{t+1})}{\delta_0^{t+1}(UCS_t, VA_t)}\right]^{\frac{1}{2}} \tag{LP7}$$

$$\psi_0\left(VA_{t+1}, AP_{t+1}, VA_t, AP_t\right) = \left[\frac{\delta_0^t(AP_{t+1}, VA_{t+1})}{\delta_0^t(AP_t, VA_t)} \times \frac{\delta_0^{t+1}(AP_{t+1}, VA_{t+1})}{\delta_0^{t+1}(AP_t, VA_t)}\right]^{\frac{1}{2}} \tag{LP8}$$

$$\psi_0\left(VA_{t+1}, TI_{t+1}, VA_t, TI_t\right) = \left[\frac{\delta_0^t(TI_{t+1}, VA_{t+1})}{\delta_0^t(TI_t, VA_t)} \times \frac{\delta_0^{t+1}(TI_{t+1}, VA_{t+1})}{\delta_0^{t+1}(TI_t, VA_t)}\right]^{\frac{1}{2}} \tag{LP9}$$

$$\psi_0\left(VA_{t+1}, CPFI_{t+1}, VA_t, CPFI_t\right) = \left[\frac{\delta_0^t(CPFI_{t+1}, VA_{t+1})}{\delta_0^t(CPFI_t, VA_t)} \times \frac{\delta_0^{t+1}(CPFI_{t+1}, VA_{t+1})}{\delta_0^{t+1}(CPFI_t, VA_t)}\right]^{\frac{1}{2}} \tag{LP10}$$

LP 6–10 represent the Bioeconomy productivity at the points $(LU_{t+1}, VA_{t+1}), (UCS_{t+1}, VA_{t+1}), (AP_{t+1}, VA_{t+1}), (TI_{t+1}, VA_{t+1})$, and $(CPFI_{t+1}, VA_{t+1})$ relative to the bioeconomy production points $(VA_t, LU_t), (VA_t, UCS_t), (VA_t, AP_t), (VA_t, TI_t), (VA_t, CPFI_t)$. To measure productivity, values from 0 to 1 are considered. It will be understood that there is a growth in TFP when the value is 1 from period t to period t+1. The bioeconomics that obtained values greater than one, then we affirm that there was growth, and if it is less, it indicates decrease. If the bioeconomy obtained a value equal to one, it will be explained that it was indifferent to the economic policies represented by the controlled variables [23].

$\delta_0^t$ and $\delta_0^{t+1}$ They represent the distance to be measured for each previously indicated productivity point for each input both for year 0 which for this study is 2012 (t) and year t+1 (2013 to 2021) respectively.

For the VRS model the condition is:

$$[\delta_0^t(LU_t UCS_t AP_t TI_t CPFI_t, VA_t)]^{-1} = max_{\phi,\lambda}\phi,$$

$$s.a - \phi VA_{it} + Y_t\lambda \geq 0,$$

$$(LU, UCS, AP, TI, CPFI)_{it} - P_t\lambda \geq 0,$$

$$\lambda \geq 0, \qquad\qquad (\text{LP11})$$

The next LPs problems are a simple variation of this:

$$[\delta_0^t(LU_t UCS_t AP_t TI_t CPFI_t, VA_t)]^{-1} = max_{\phi,\lambda}\phi,$$

$$s.a - \phi VA_{i,t+1} + Y_{t+1}\lambda \geq 0,$$

$$((LU, UCS, AP, TI, CPFI)_{i,t+1} - P_{t+1}\lambda \geq 0,$$

$$\lambda \geq 0, \qquad\qquad (\text{LP3})$$

$$[\delta_0^t((LU, UCS, AP, TI, CPFI))_{t+1}, VA_{t+1}]^{-1} = max_{\phi,\lambda}\phi,$$

$$s.a - \phi VA_{i,t+1} + Y_t\lambda \geq 0,$$

$$((LU, UCS, AP, TI, CPFI))_{it} - P_t\lambda \geq 0,$$

$$\lambda \geq 0, \qquad\qquad (\text{LP12})$$

$$[\delta_0^{t+1}(LU_t UCS_t AP_t TI_t CPFI_t)_t, VA_t]^{-1} = max_{\phi,\lambda}\phi,$$

$$s.a - \phi \varrho VA_{it} + Y_t\lambda \geq 0,$$

$$(LU_t UCS_t AP_t TI_t CPFI_t)_{it} - P_{t+1}\lambda \geq 0,$$

$$\lambda \geq 0, \qquad\qquad (\text{LP13})$$

Following Farrel [14], the LPs 12 and 13 are points of comparable bioeconomics with different technologies in changed types of periods, so the parameter $\phi$ should not be greater than one, as explained above [23].

In LP 13 is noted that the period t1 in the bioeconomy point is associated with the technology of the period t+1. In the bioeconomy of developed countries it is possible to find $\phi$ values less than 1, when technology has decreased. However, in LP 5 it is observed that growth is possible, that is, greater than 1.

Finally, the ϕ, and λ are likely to take different values within 10–13 LPs, because it represents values for each region in the sample.

### 3.1 Control variables and panel data sources

Value Agriculture ($VA_{it}$) represents the Output where the bioeconomy (residual biomass) is integrated. Crop and livestock statistics are recorded for 278 products, covering the following categories: 1) crops primary, 2) crops processed, 3) live animals, and 4) livestock primary [19,25,26].

Land use ($LU_{it}$) signifies input 1, Land use refers to the socio-economic use of land (e.g. agriculture, forestry, recreation, and housing). In particular, it defines various services such as agriculture, forestry, industry, transportation, housing, and other services that use the land as a natural and/or economic resource. This variable can influence the level of innovation and help improve his TFP in the bioeconomy [27,28].

Unit Capital Stock ($UCS_{it}$) denotes input 2, which can affect technological innovation changes. It can also be described as the difference between gross capital stock and consumption of fixed capital [29,30].

Annual population ($AP_{it}$) represents input 3, which can affect the changes in technical efficiency, in conditions of Covid-19 affecting the working market [31,32].

Trade Indices ($TI_{it}$) represents input 4, which can affect technological innovation changes. It includes re-exports. According to the FAO methodology, the quantity of food and agricultural exports included in the FAOSTAT database is expressed in terms of weight (tonnes) for all commodities except for live animals which are expressed in units (heads); poultry, rabbits, pigeons and other birds are expressed in thousand units [28,33].

Consumer Prices, Food Indices (2015 = 100) ($CPFI_{it}$) represents input 5, which can affect technological innovation changes. Consumer Price Indices measure the price change between the current and reference periods of an average basket of goods and services purchased by households [27,31].

This article used panel data for seven regions of the world: Northern America, Central America, South America, Northern Europe, Southern Europe, and Western Europe from FAO Statistic (RRID: SCR_006914), and CIA Factbook Resource ID: SCR_023548 see Figs 3 and 4. The specific descriptive statistics are shown in Table 1. The full protocol can be found on protocols.io (Fig 5).

Table 2 describes the statistics of the data used [35]. On average, biobased economies present an added value (Output VAit) of 177.3 billion dollars, with a standard deviation of 121.6 billion dollars. Of the five controlled inputs, land use (LUit) is an important variable to assess

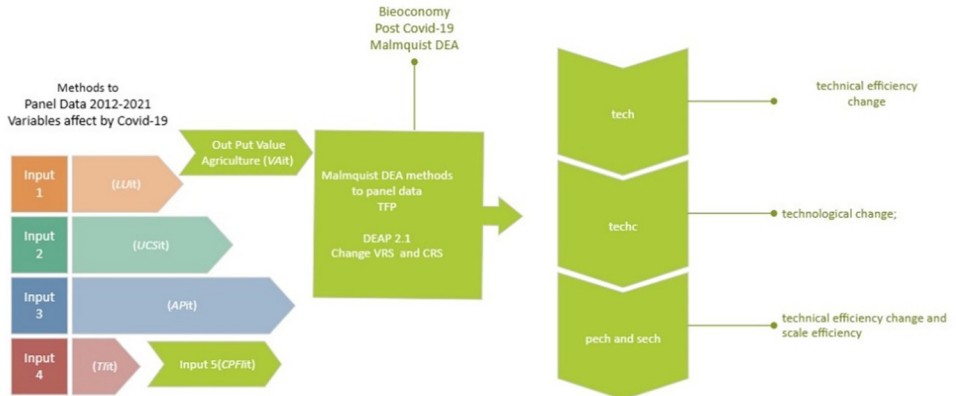

**Fig 3. Map of last 28-day cases of Covid-19 03/10/2023.** Source: Dong, Du, and Gardner [34].

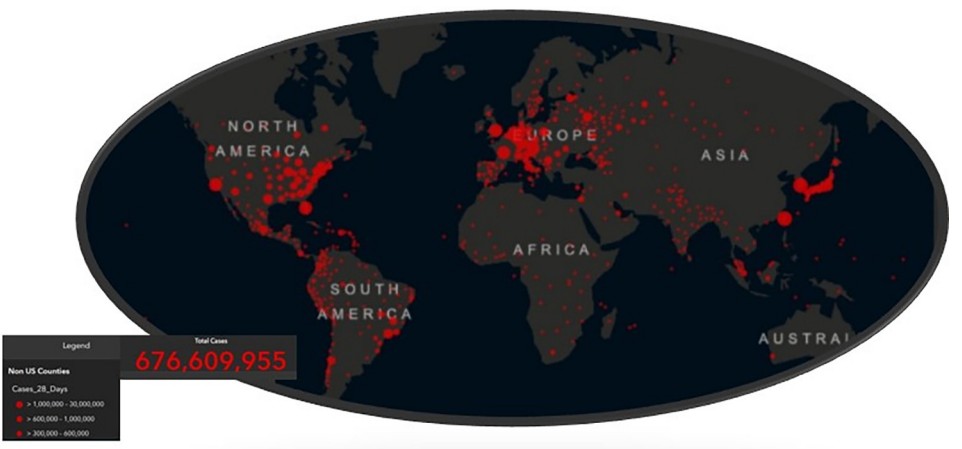

**Fig 4. Map of world regions source: CIA Factbook—(Maps—The World Factbook).**

the level of innovation. On average, 211011.9 ha were affected with a standard deviation of 2081199.3 ha. Input 2 affects technology because it has to do with the stock of capital units, in this sense it is observed that on average 330613.2 units were affected with a standard deviation of 204580.7. Input 3 is the population growth rate considered in this model because it refers to technical efficiency. On average 233,731 thousand people grew per year with a standard deviation of 116,894 thousand people. Input 4 has to do with trade indices, a variable affected during Covid. On average, the index was around 105.12 with a standard deviation of 10.85. And finally input 5 is the consumer price index. On average, the increase in prices was 109.37 and a standard deviation of 26.54.

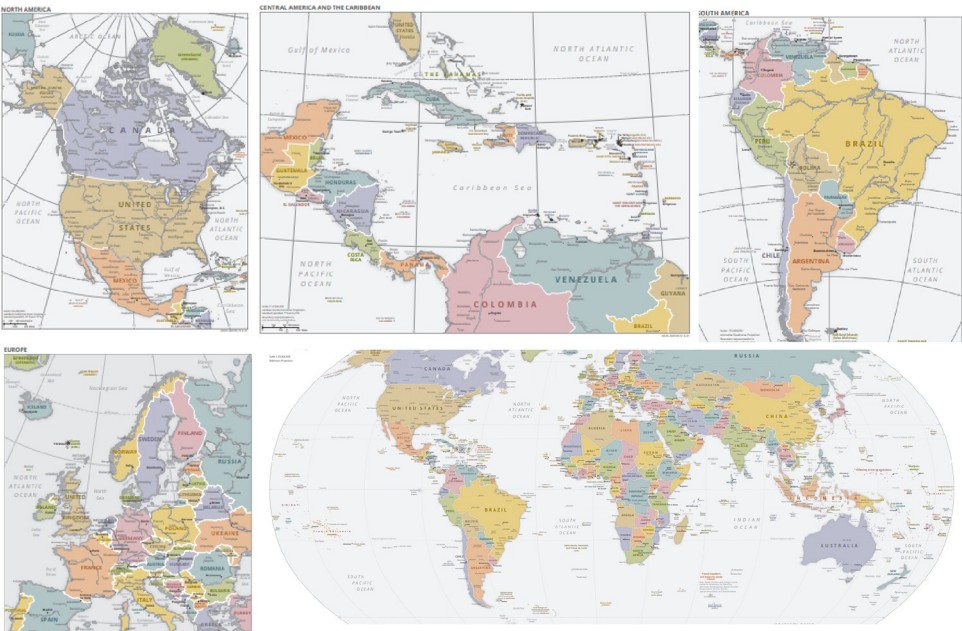

**Fig 5. Methodological Diagram Indices Malmquist DEA.**

**Table 2. Statistical descriptive variables.**

|  | Value Agriculture | Land use | Unit Capital Stock | Annual population | Trade Indices | Consumer Prices, Food Indices (2015 = 100) |
|---|---|---|---|---|---|---|
| Unit | 1000 US$ | 1000 ha | millions $ | 1000 persons | Index | Indices |
| Abbreviation Variable | (VA) | (LU) | (UCS) | (AP) | (TI) | (CPFI) |
| OBS PANEL DATA | 60 | 60 | 60 | 60 | 60 | 60 |
| Mean | 177338586 | 211011.9 | 330613.2 | 233731 | 105.12 | 109.37 |
| Stándar Desviation | 121656260 | 208199.3 | 204580.7 | 116894 | 10.85 | 26.54 |
| Mínimum | 54796241 | 38304.1 | 43524.9 | 101015 | 83 | 81.04 |
| Maximum | 389720489 | 542581.8 | 741511.1 | 434254 | 131 | 258.61 |
| Confidence Level (95.0%) | 31427144.5 | 53783.6 | 52848.8 | 30197 | 2.8 | 6.86 |

## IV. Results and discussion

This investigation was centered on measuring the TFP bioeconomy post-Covid-19 in six areas of the world. For this the application of Malmquist DEA methods to panel data calculated indices of total factor productivity (TFP) change to measure four indices; technological change; technical efficiency change and scale efficiency change [23,36].

Table 3 shows the Malmquist index summary by region. In general, during the study period (2012–2021) the change in the growth rate of total factor productivity (tfpch) was as expected due to a decrease of 12%. This decrease was mainly due to the decrease in technical efficiency (11.7%). Bio-based economies maintained their technologies (techch) at their frontier production level. Due to the decrease in technical efficiency, the most affected regions were Northern America with a decrease of 21.6% and Southern Europe at 10.15, and Western Europe at 11.7%. This trend is similar and coincides with works by other authors [37–39].

Table 4 shows the Malmquist index summary by annual mean. It is noted that the bio-based economies experienced a decreasing change in TFP, however, in the inter-annual analysis it is observed that this trend was growing and improving until reaching the year 2019 when Covid 19 occurs, the fall is manifested or maintained in 2020, and starting in 2021 a trend to improve and improve bio-based economies. It is noteworthy, as indicated by other studies [30,40–43], in this new context the new strategies and innovation initiatives in the bio-based economies are considered, This situation varied a little in the efficiency of production at scale (sech), noting a decrease, although its changes are invariable before the situation. Some authors indicate that these processes in terms of technology, circular bioeconomy, and wastewater treatment, contributed and contribute in terms of economic policies in the country agenda [44], however as shown in the map in Figs 2 and 6 there is an uneven development in the subject studied [45–48].

**Table 3. Malmquist Index of annual means 2012–2021.**

| Region | effch | techch | pech | sech | tfpch |
|---|---|---|---|---|---|
| Northern America | 1 | 0.784 | 1 | 1 | 0.784 |
| Central America | 1 | 0.911 | 1 | 1 | 0.911 |
| South America | 1.005 | 0.907 | 1.005 | 1 | 0.911 |
| Northern Europe | 1 | 0.92 | 1 | 1 | 0.92 |
| Southern Europe | 1 | 0.899 | 1 | 1 | 0.899 |
| Western Europe | 1 | 0.883 | 1 | 1 | 0.883 |
| mean | 1.001 | 0.883 | 1.001 | 1 | 0.884 |

[Note that all Malmquist index averages ares geometric means].

**Table 4. Malmquist index of annual means variation 2012–2021.**

| years | effch | techch | pech | sech | tfpch |
|---|---|---|---|---|---|
| 2013 | 0.996 | 0.716 | 0.996 | 1 | 0.713 |
| 2014 | 0.996 | 0.82 | 0.996 | 1 | 0.817 |
| 2015 | 0.997 | 0.87 | 0.997 | 1 | 0.867 |
| 2016 | 0.997 | 0.9 | 0.997 | 1 | 0.898 |
| 2017 | 0.998 | 0.915 | 0.998 | 1 | 0.913 |
| 2018 | 1 | 0.932 | 1 | 1 | 0.932 |
| 2019 | 0.917 | 1.02 | 0.917 | 1 | 0.936 |
| 2020 | 1.109 | 0.834 | 1.123 | 0.987 | 0.925 |
| 2021 | 1.007 | 0.976 | 0.994 | 1.013 | 0.982 |
| mean | 1.001 | 0.883 | 1.001 | 1 | 0.884 |

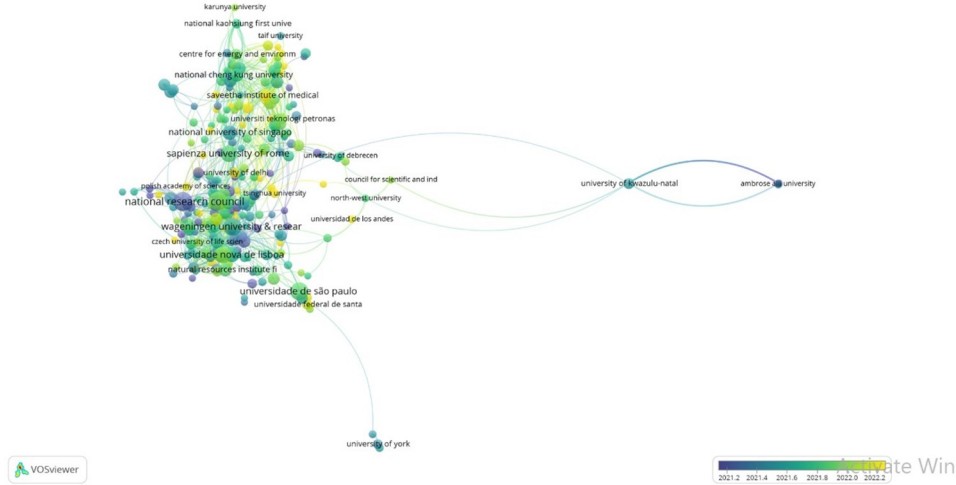

**Fig 6. Map based on academic institution Dimension data for Bioeconomy during and post Covid-19.**

**Table 5. Measure mean constant return scale technical efficiency relation to technology year 2012–2021.**

| year | crs te rel to tech in yr | | | vrs |
|---|---|---|---|---|
| | ********************** | | | |
| | Last year (t-1) | Current year (t) | Next year (t+1) | |
| 2012 | 0 | 0.818 | 1.358 | 0.818 |
| 2013 | 0.76 | 0.815 | 1.089 | 0.815 |
| 2014 | 0.775 | 0.813 | 0.993 | 0.813 |
| 2015 | 0.783 | 0.811 | 0.945 | 0.811 |
| 2016 | 0.792 | 0.809 | 0.92 | 0.809 |
| 2017 | 0.791 | 0.808 | 0.897 | 0.808 |
| 2018 | 0.801 | 0.808 | 0.811 | 0.808 |
| 2019 | 0.812 | 0.741 | 0.916 | 0.741 |
| 2020 | 0.707 | 0.81 | 0.849 | 0.812 |
| 2021 | 0.818 | 0.824 | 0 | 0.824 |

[Note that t-1 in year 1 and t+1 in the final year are not defined].

**Table 6. Measure summary constant return scale technical efficiency relation to technology by year and region 2012–2021.**

| crs te rel to tech in yr | Years | Northern America | Central America | South America | Northern Europe | Southern Europe | Western Europe |
|---|---|---|---|---|---|---|---|
| Last yer (t-1) | 2012 | 0 | 0 | 0 | 0 | 0 | 0 |
| | 2013 | 0.1 | 0.999 | 0.534 | 1.006 | 0.918 | 1.001 |
| | 2014 | 0.115 | 0.994 | 0.607 | 1.005 | 0.928 | 1.001 |
| | 2015 | 0.125 | 0.99 | 0.633 | 1.001 | 0.949 | 1.003 |
| | 2016 | 0.133 | 0.991 | 0.635 | 1.021 | 0.966 | 1.007 |
| | 2017 | 0.139 | 0.998 | 0.638 | 0.997 | 0.975 | 1.002 |
| | 2018 | 0.143 | 0.993 | 0.64 | 1.051 | 0.977 | 1.005 |
| | 2019 | 0.146 | 0.993 | 0.645 | 0.668 | 1.243 | 1.177 |
| | 2020 | 0.148 | 0.991 | 0.552 | 0.811 | 0.811 | 0.931 |
| | 2021 | 0.167 | 1.002 | 0.744 | 1.01 | 0.977 | 1.005 |
| Current year (t) | 2012 | 0.167 | 1 | 0.744 | 1 | 1 | 1 |
| | 2013 | 0.167 | 1 | 0.726 | 1 | 1 | 1 |
| | 2014 | 0.167 | 1 | 0.71 | 1 | 1 | 1 |
| | 2015 | 0.167 | 1 | 0.696 | 1 | 1 | 1 |
| | 2016 | 0.167 | 1 | 0.685 | 1 | 1 | 1 |
| | 2017 | 0.167 | 1 | 0.679 | 1 | 1 | 1 |
| | 2018 | 0.167 | 1 | 0.679 | 1 | 1 | 1 |
| | 2019 | 0.167 | 1 | 0.572 | 0.706 | 1 | 1 |
| | 2020 | 0.185 | 1 | 0.675 | 1 | 1 | 1 |
| | 2021 | 0.167 | 1 | 0.779 | 1 | 1 | 1 |
| Next year (t+1) | 2012 | 0.333 | 1.611 | 1.03 | 1.509 | 1.667 | 2 |
| | 2013 | 0.25 | 1.337 | 0.857 | 1.319 | 1.272 | 1.5 |
| | 2014 | 0.222 | 1.23 | 0.783 | 1.198 | 1.19 | 1.333 |
| | 2015 | 0.208 | 1.176 | 0.747 | 1.138 | 1.15 | 1.25 |
| | 2016 | 0.2 | 1.141 | 0.724 | 1.134 | 1.123 | 1.2 |
| | 2017 | 0.194 | 1.119 | 0.717 | 1.081 | 1.106 | 1.167 |
| | 2018 | 0.19 | 1.09 | 0.594 | 0.9 | 0.952 | 1.143 |
| | 2019 | 0.208 | 1.089 | 0.703 | 0.833 | 1.345 | 1.317 |
| | 2020 | 0.185 | 1.073 | 0.703 | 1.062 | 1.07 | 1.001 |
| | 2021 | 0 | 0 | 0 | 0 | 0 | 0 |
| Variable to return scale (vrs) | 2012 | 0.167 | 1 | 0.744 | 1 | 1 | 1 |
| | 2013 | 0.167 | 1 | 0.726 | 1 | 1 | 1 |
| | 2014 | 0.167 | 1 | 0.71 | 1 | 1 | 1 |
| | 2015 | 0.167 | 1 | 0.696 | 1 | 1 | 1 |
| | 2016 | 0.167 | 1 | 0.685 | 1 | 1 | 1 |
| | 2017 | 0.167 | 1 | 0.679 | 1 | 1 | 1 |
| | 2018 | 0.167 | 1 | 0.679 | 1 | 1 | 1 |
| | 2019 | 0.167 | 1 | 0.572 | 0.706 | 1 | 1 |
| | 2020 | 0.2 | 1 | 0.675 | 1 | 1 | 1 |
| | 2021 | 0.167 | 1 | 0.779 | 1 | 1 | 1 |

Table 5 shows the measure of distances mean of Constant Returns to Scales (CRS) technical efficiency and your relation with the technology. It is important to assess this type of relationship given that under Covid conditions it was evidenced that the decreasing changes in TFP were mainly due to changes in technical efficiency (EFFCH), this has to do with the people affected by Covid having an impact on the availability of labor in companies. Some authors highlight this importance in the Covid period [49–54].

It is noted that the economies of the regions studied in constant yields from the previous year (t-1) to the current time (t) and the subsequent one (t+1) show a tendency to grow and it is noticeable how it decreased in the year 2019 and then it resumes the growth trend, although, at present, it tends to reduce its inter-annual changes as explained in Table 3. On the other hand, in the variable scale yields an increasing trend was observed that decreases in the year of Covid but resumed its trend in subsequent years. Recent research characterizes this trend as the use of an emerging technology that is necessary for policymakers to consider in the country [55–57].

Table 6 shows the technical efficiency related to technology by year and region, 2012–2021.

It is noted that the inter-annual variations were more favorable in variable yields than constant yields. Note that in the period that the North American and South American regions presented decreasing returns in variable return scale (VRS), on the other hand, in constant return scale (CRS) conditions, the European regions maintained rational growth. These results were congruent with the map of Figs 2 and 6, where the authors and universities or academics institution highlight this topic of research, this situation has to do with the policies and government measures prioritized in the research topics with a Bioeconomy approach as exposed by the research in the study areas in Europe. Fig 6 shows the institution that has been contributing to bioeconomy as an alternative for Covid-19, 2,344 institutions were found, but only 263 worked on this topic [46,58–62].

## V. Conclusion

This article evaluated the indices of the Malmquist DEA methods for panel data. These indices were the change in total factor productivity (TFP); technological change; technical efficiency change and scale efficiency change. The results showed a downward trend affected in 2019 by Covid, however, it was possible to recover in the following year. One of the conclusions of these results is the effect of the immediate strategies implemented by the governments of the region. This effect was slightly slower in the North American, Southeast and Eastern European regions. The bias in which governments act to solve post-Covid-19 economic problems is notorious. This situation is evidenced in the maps of authors, countries and academic institutions that address the issue of the Bioeconomy as an alternative to face the consequences of the pandemic. The variables studied are macroeconomic and were expressed in the economic policies of each country, so the results showed the bias in the measures applied and the results in the economic dynamics of their respective countries.

Finally, it is concluded that the measures implemented by the governments in the studied regions had an increasing effect in conditions of variable scale returns. In other words, the companies that remained on a constant scale decreased or were not productive or technically efficient.

## Author Contributions

**Conceptualization:** C. A. Zuniga-Gonzalez.

**Data curation:** C. A. Zuniga-Gonzalez.

**Formal analysis:** C. A. Zuniga-Gonzalez.

**Funding acquisition:** C. A. Zuniga-Gonzalez.

**Investigation:** C. A. Zuniga-Gonzalez.

**Methodology:** C. A. Zuniga-Gonzalez.

**Project administration:** C. A. Zuniga-Gonzalez.

**Resources:** C. A. Zuniga-Gonzalez.

**Software:** C. A. Zuniga-Gonzalez.

**Supervision:** C. A. Zuniga-Gonzalez.

**Validation:** C. A. Zuniga-Gonzalez.

**Visualization:** C. A. Zuniga-Gonzalez.

**Writing – original draft:** C. A. Zuniga-Gonzalez.

**Writing – review & editing:** C. A. Zuniga-Gonzalez.

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
