## [Editor Report · Decision Letter 0]

24 Apr 2023

PONE-D-23-10369

TFP Bioeconomy Impact post Covid-19 on the agricultural economy

PLOS ONE

Dear Dr. Zúniga-González,

Thank you for submitting your manuscript to PLOS ONE. After careful consideration, we feel that it has merit but does not fully meet PLOS ONE’s publication criteria as it currently stands. Therefore, we invite you to submit a revised version of the manuscript that addresses the points raised during the review process.

The manuscript requires important changes:

1. Add a map of the regions: Northern America, Central America, South America, Northern Europe, Southern Europe, and Western Europe with the latest data on the economic impact of COVID in those regions

2. In the Literature Review section you must make a flow chart or incorporate VosViewer to strengthen the section, mainly the text: woźniak & tyczewska [15] presents the challenges and threats during the covid-19 pandemic, as well as opportunities that can be broucht by for bioeconomy Development and the author must use it for discussion

3. Add the objective, innovation, scope, research limitations and a methodological diagram

4. Tables 1-5 must be built by the author

5. Figure 1 should be improved

6. The conclusions are very poor, the agricultural sustainability approach and its prospective for decision making is not discussed

We look forward to receiving your revised manuscript.

Kind regards,

Noé Aguilar-Rivera

Academic Editor

PLOS ONE

---

## [Author Response · Author response to Decision Letter 0]

5 May 2023

Maidelyn R. Peregrin

Editor Academico

Dear editor thanks for all observations. 

Dear editor, thank you for all your observations that surely contribute to improving the quality of the research. Below I present in detail each of the improvements incorporated.

1. We note that Figures 4 & 5 in your submission contain map/satellite images which may be copyrighted. 

The Figure 5 was change as you suggested, regarding the Figure 4 is cited and was change They only ask that cite to Source: Dong, Du, and Gardner (2020). Was selected the Maps at the CIA (public domain): https://www.cia.gov/library/publications/the-world-factbook/index.html and https://www.cia.gov/library/publications/cia-maps-publications/index.html

2. Can you please upload an additional copy of your revised manuscript that does not contain any tracked changes or highlighting as your main article file. This will be used in the production process if your manuscript is accepted. Please amend the file type for the file showing your changes to Revised Manuscript w/tracked changes. Please follow this link for more information: http://blogs.PLOS.org/everyone/2011/05/10/how-to-submit-your-revised-manuscript/

Yes is was made as you indicated.

3. Please include a legend/caption for figures 2, 3, 4, 5, and 6 in your main document.

Dear regarding the type of Map if not necessary to aggregate a Legend. This type of map 

Bst Rgs

Carlos

---

## [Decision Letter · Decision Letter 1]

2 Jun 2023

PONE-D-23-10369R1

TFP Bioeconomy Impact post Covid-19 on the agricultural economy

PLOS ONE

Dear Dr. Zúniga-González,

Thank you for submitting your manuscript to PLOS ONE. After careful consideration, we feel that it has merit but does not fully meet PLOS ONE’s publication criteria as it currently stands. Therefore, we invite you to submit a revised version of the manuscript that addresses the points raised during the review process.

We look forward to receiving your revised manuscript.

Kind regards,

Noé Aguilar-Rivera

Academic Editor

PLOS ONE

Reviewers' comments:

Reviewer's Responses to Questions

**Comments to the Author**

1. If the authors have adequately addressed your comments raised in a previous round of review and you feel that this manuscript is now acceptable for publication, you may indicate that here to bypass the “Comments to the Author” section, enter your conflict of interest statement in the “Confidential to Editor” section, and submit your "Accept" recommendation.

Reviewer #1: (No Response)

Reviewer #2: (No Response)

2. Is the manuscript technically sound, and do the data support the conclusions?

Reviewer #1: Partly

Reviewer #2: Yes

3. Has the statistical analysis been performed appropriately and rigorously? 

Reviewer #1: Yes

Reviewer #2: No

4. Have the authors made all data underlying the findings in their manuscript fully available?

Reviewer #1: Yes

Reviewer #2: Yes

5. Is the manuscript presented in an intelligible fashion and written in standard English?

Reviewer #1: No

Reviewer #2: No

6. Review Comments to the Author

Reviewer #1: The author used panel date to investigate the impact of Covid-19 on bio-economy from six regions in the world, I think the aim of this paper is worth looking into, however, this paper needs tremendous polishing before it could meet publication requirement.

1.Language needs to be polished by an native English professional. This paper reads ambiguously and redundantly.

2.In the second line of the introduction, the author had put “Some authors have pointed out” without citing the work you were referring to.

3.In the forth paragraph of the introduction, the authors put “so the contribution of this study is focused on measuring the change in TFP in six regions of the world. Part of the research efforts” . From my perspective, the authors were trying to put forward the significance of the paper, however, you did not explain clearly why and how this method and this outcome would serve these means.

4.By the end of this paragraph, there was an extra “That”

5.The last paragraph of the introduction, the authors seemed to left out the number this paper was divided into, therefore this paragraph was a little confusing. The authors should treat this submission more seriously and avoid such mistakes.

6.I would suggest the authors reorganize the introduction part, explain clearly the background of the research, the methods you were using and why is was appropriate, the aim and objectives of this paper, and what outcome and achievement you were expecting from this paper.

7.I was confused by the way the authors arranged this paper, as you used Roman numbers for the introduction and literature review section but there was no numbering for total factor productivity. I suggest you rearrange each section and make it clear.

8. Even though the authors have listed many related papers in the literature review section, the discussion was absent as you went directly from literature review to total factor productivity. Literature review section was for you to list out related works and comment on the research background, then it leads to the research blank that this paper should be fulfilling. Without the discussion, the literature review is not complete.

9.Methods and data should be merged into one big section, I suggestion the authors state clearly why this method was employed first, then construct your model, then explain the data source. I don’t understand why controlled variables were not explain along with the model you constructed.

10.For statistics, I suggest the authors put a table of inputs and outputs and then list the statistics for discussion.

11.The authors had discussed the results from your data, however, you did not form an discussion with existing literatures, I suggest you add a “discussion” section to compare your research findings with existing research and discussion the significance and difference in your findings.

12.The aims of this paper is meaningful, therefore I look forward to see the finely polished version of this paper.

Reviewer #2: Although I think the paper addresses a very relevant theme, the way it is presented at the moment makes it no publishable in my opinion. The paper lacks a proper introduction and an adequate analysis of the current literature. The authors provide a meta-analysis but it does not explain in which stream of the literature it is positioned and what the paper adds new and relevant to that stream of literature. The empirical work "seems" very interesting but it needs better explaination and presentation of data and results. Figures do not help much in the understanding of the problem addressed. The conclusions needs to be totally rethought of and re-written

7. PLOS authors have the option to publish the peer review history of their article (what does this mean?). If published, this will include your full peer review and any attached files.

Reviewer #1: No

Reviewer #2: No

---

## [Author Response · Author response to Decision Letter 1]

14 Jun 2023

6/10/2023

Noé Aguilar-Rivera

Editor Academico

Dear editor thanks for all observations 

Dear Editor, thank you for all your observations that surely contribute to improving the quality of the research. Below I present in detail each of the improvements incorporated. All the change made go in yellow color.

The author used panel data to investigate the impact of Covid-19 on the bio-economy from six regions in the world, I think the aim of this paper is worth looking into, however, this paper needs tremendous polishing before it could meet publication requirements.

I appreciate your comments. The panel data is an essential condition for the Malmquist indices. I agree to make the results more visible in the face of regionalization and the measurement of productivity and efficiency that after Covid the governments justified with their respective economic policies.

1. Language needs to be polished by a native English professional. This paper reads ambiguously and redundantly.

A review was done with a native English colleague, I hope your valuable observations in this regard are improved.

2. In the second line of the introduction, the author had put “Some authors have pointed out” without citing the work you were referring to. 

It was added.

3. In the fourth paragraph of the introduction, the authors put “so the contribution of this study is focused on measuring the change in TFP in six regions of the world. Part of the research efforts”. From my perspective, the authors were trying to put forward the significance of the paper, however, you did not explain clearly why and how this method and this outcome would serve these means.

Thanks for this observation. It was eliminate and added in the fifth paragraph, I saw necessary separate in the next paragraph. This paragraph was reword. 

4. By the end of this paragraph, there was an extra “That”

It was eliminate

5. In The last paragraph of the introduction, the authors seemed to left out the number this paper was divided into, therefore this paragraph was a little confusing. The authors should treat this submission more seriously and avoid such mistakes.

Thanks for this observation. It was correct. 

6. I would suggest the authors reorganize the introduction part, explain clearly the background of the research, the methods you were using and why is was appropriate, the aim and objectives of this paper, and what outcome and achievement you were expecting from this paper. 

Thanks for this observation. It was added the table3 that resume the TFP used in bio based economic. The section was re organized and clarify the background, aim, contribution and objective.

7. I was confused by the way the authors arranged this paper, as you used Roman numbers for the introduction and literature review section but there was no numbering for total factor productivity. I suggest you rearrange each section and make it clear.

My sincerest apologies for that. That was duly numbered both by section and by subsections.

8. Even though the authors have listed many related papers in the literature review section, the discussion was absent as you went directly from literature review to total factor productivity. Literature review section was for you to list out related works and comment on the research background, then it leads to the research blank that this paper should be fulfilling. Without the discussion, the literature review is not complete. 

I also apologize for this observation. The literature review focused on two components: one was the bioeconomy and the other was total factor productivity. One is the theoretical approach towards which we want to measure and the other is the tool that will measure it. Make an effort to bring these two components together in the discussion you recommend. Thanks for this observation.

9. Methods and data should be merged into one big section, I suggestion the authors state clearly why this method was employed first, then construct your model, then explain the data source. I don’t understand why controlled variables were not explain along with the model you constructed. 

The model was contructed and explained.

It was made. 3. 1 3.1 Control Variables and Panel Data Sources 

10. For statistics, I suggest the authors put a table of inputs and outputs and then list the statistics for discussion. 

It was added.

11. The authors had discussed the results from your data, however, you did not form an discussion with existing literatures, I suggest you add a “discussion” section to compare your research findings with existing research and discussion the significance and difference in your findings.

Yes it was made. It was added.

12. The aims of this paper is meaningful, therefore I look forward to see the finely polished version of this paper. 

Thank you for your comment.

---

## [Decision Letter · Decision Letter 2]

20 Jun 2023

PONE-D-23-10369R2TFP Bioeconomy Impact post Covid-19 on the agricultural economyPLOS ONE

Dear Dr. Zúniga-González,

Thank you for submitting your manuscript to PLOS ONE. After careful consideration, we feel that it has merit but does not fully meet PLOS ONE’s publication criteria as it currently stands. Therefore, we invite you to submit a revised version of the manuscript that addresses the points raised during the review process.

We look forward to receiving your revised manuscript.

Kind regards,

Noé Aguilar-Rivera

Academic Editor

PLOS ONE

Journal Requirements:

Reviewers' comments:

Reviewer's Responses to Questions

**Comments to the Author**

1. If the authors have adequately addressed your comments raised in a previous round of review and you feel that this manuscript is now acceptable for publication, you may indicate that here to bypass the “Comments to the Author” section, enter your conflict of interest statement in the “Confidential to Editor” section, and submit your "Accept" recommendation.

Reviewer #1: (No Response)

Reviewer #2: All comments have been addressed

2. Is the manuscript technically sound, and do the data support the conclusions?

Reviewer #1: Yes

Reviewer #2: Partly

3. Has the statistical analysis been performed appropriately and rigorously? 

Reviewer #1: Yes

Reviewer #2: Yes

4. Have the authors made all data underlying the findings in their manuscript fully available?

Reviewer #1: Yes

Reviewer #2: Yes

5. Is the manuscript presented in an intelligible fashion and written in standard English?

Reviewer #1: Yes

Reviewer #2: Yes

6. Review Comments to the Author

Reviewer #1: The language has been improved significantly, as well as the quality of the paper. However, there are still some shortcoming I think the authors should work on：

1. The literature review section still lacks review and discussion.

2. You need to discuss the descriptive statistics before starting analysing.

3. Form a discussion with current literature by comparing your results to other research before you draw the conclusions.

I look forward to see the three sections added before this paper goes to publication.

Reviewer #2: Thank you for the new version of the paper which has beenmeaningfully improved. I still ave probems with the literature review which seems to me quite random and it is not very clear how and to what extent your work contribute to that strand of literature.

I find fig. 4 and fig. 5 a bit useless and fig. 6 quite unreadable.

I also suggest to imoprove the conclusions to highlight your original contribution and addition to the existing literature status quo.

7. PLOS authors have the option to publish the peer review history of their article (what does this mean?). If published, this will include your full peer review and any attached files.

Reviewer #1: No

Reviewer #2: No

---

## [Author Response · Author response to Decision Letter 2]

4 Jul 2023

2023/07/04

Noé Aguilar-Rivera

Academic Editor

PLOS ONE

Dear academic editor, thank you for your valuable management in this editorial process. Below I present the rebuttal letter indicating each of the improvements to the reviewers' observations.

Comments to the Author

1. If the authors have adequately addressed your comments raised in a previous round of review and you feel that this manuscript is now acceptable for publication, you may indicate that here to bypass the “Comments to the Author” section, enter your conflict of interest statement in the “Confidential to Editor” section, and submit your "Accept" recommendation.

Reviewer #1: (No Response)

Reviewer #2: All comments have been addressed

2. Is the manuscript technically sound, and do the data support the conclusions?

Reviewer #1: Yes

Reviewer #2: Partly

3. Has the statistical analysis been performed appropriately and rigorously?

Reviewer #1: Yes

Reviewer #2: Yes

4. Have the authors made all data underlying the findings in their manuscript fully available?

Reviewer #1: Yes

Reviewer #2: Yes

5. Is the manuscript presented in an intelligible fashion and written in standard English?

Reviewer #1: Yes

Reviewer #2: Yes

6. Review Comments to the Author

Reviewer #1: The language has been improved significantly, as well as the quality of the paper. However, there are still some shortcoming I think the authors should work on：

1. The literature review section still lacks review and discussion.

I appreciate the reviewer's comments.Corrections were made, marked in yellow. A paragraph was added at the beginning of section 2.1 and at the end of section 2.3, a paragraph was added summarizing the contributions of research on the subject of bioeconomy, and the Malmquist indices tool to measure efficiency and productivity. of the bioeconomy during and post covid. The emptiness is made clear and hence the contribution of this work.

2. You need to discuss the descriptive statistics before starting analyzing.

We appreciate the reviewer's comments. The discussion of Table 1 moved to the results and discussion section before starting the analysis. It Added the discussion. 

3. Form a discussion with current literature by comparing your results to other research before you draw the conclusions.

I look forward to see the three sections added before this paper goes to publication.

Thanks for your observation, the tree section were added. 

Reviewer #2: Thank you for the new version of the paper which has beenmeaningfully improved. I still have probems with the literature review, which seems to me quite random and it is not very clear how and to what extent your work contribute to that strand of literature.

Dear thanks for these observations. Two paragraph has added with the intention of combine the bioeconomy with the índices Malmquist tool. The use of Indices Malmquist as measure tool of the Bioeconomy is a contribution study with controlled variables to level regional decreased or were not productive or technically efficient.

I find fig. 4 and fig. 5 a bit useless and fig. 6 quite unreadable.

Regarding the fig. 4 is for indicating region Covid affectation. The fig 4 is for indicating region of the study both suggested by back reviewer. Fig 5 is with the standard as journal ask. 

Regarding the Fig 6 is the better resolution on The VOSviewer software (RRID: SCR_023516) it also, was suggested by other back reviewer. 

I also suggest to improve the conclusions to highlight your original contribution and addition to the existing literature status quo.

Thanks for this information and observation. It was added and improved. 

7. PLOS authors have the option to publish the peer review history of their article (what does this mean?). If published, this will include your full peer review and any attached files.

Do you want your identity to be public for this peer review? For information about this choice, including consent withdrawal, please see our Privacy Policy.

Reviewer #1: No

Reviewer #2: No

Kind regards,

C.A. Zuniga-Gonzalez

Researcher

UNAN Leon

---

## [Editor Report · Decision Letter 3]

6 Jul 2023

TFP Bioeconomy Impact post Covid-19 on the agricultural economy

PONE-D-23-10369R3

Dear C. A. Zúniga-González

We’re pleased to inform you that your manuscript has been judged scientifically suitable for publication and will be formally accepted for publication once it meets all outstanding technical requirements.

Kind regards,

Noé Aguilar-Rivera

Academic Editor

PLOS ONE
---

## [Editor Report · Acceptance letter]

11 Jul 2023

PONE-D-23-10369R3 

TFP Bioeconomy Impact post Covid-19 on the agricultural economy 

Dear Dr. Zúniga-González:

I'm pleased to inform you that your manuscript has been deemed suitable for publication in PLOS ONE. Congratulations! Your manuscript is now with our production department. 

Kind regards, 

on behalf of

Dr. Noé Aguilar-Rivera 

Academic Editor

PLOS ONE